# The Diagnostic Process for Children with Autism Spectrum Disorder: A Preliminary Study of Jordanian Parents’ Perspectives

**DOI:** 10.3390/children10081394

**Published:** 2023-08-15

**Authors:** Mizyed Hyassat, Ahmad Al-Makahleh, Zahraa Rahahleh, Nawaf Al-Zyoud

**Affiliations:** 1Department of Special Education, Princess Rahma University College, Al-Balqa Applied University, Salt 19117, Jordan; dr.amakahleh@bau.edu.jo (A.A.-M.); zahraa.r@bau.edu.jo (Z.R.); 2Department of Psychological Sciences, College of Education, Qatar University, Doha 93001, Qatar; nawaf.alzyoud@qu.edu.qa

**Keywords:** autism spectrum disorder, diagnosis, Jordan, parents

## Abstract

Although extensive research has been conducted worldwide to investigate the diagnostic process of Autism Spectrum Disorder (ASD), Jordanian parents’ experiences have been overlooked. This study explored parents’ journeys toward receiving diagnoses for their children with ASD. In particular, it aimed to provide a clear picture of the process for obtaining these diagnoses for children in Jordan. Methods: Eighteen semi-structured interviews were carried out with 12 mothers and six fathers of children with ASD aged 5 to 11 years old. Results: The coding process was based on a thematic analysis method and resulted in the identification of three overlapping themes: dissatisfaction with professionals’ abilities to approach parents, an unstructured diagnostic process, and perspectives on diagnosis tools. Conclusions: Our data upheld the idea that parental satisfaction with the diagnostic process is influenced by the duration of the process, the information provided, the support offered, and the communication approach used by professionals. Within the local cultural context, parents were significantly impacted by the societal stigma associated with disability when they sought diagnoses for their children with ASD.

## 1. Introduction

### 1.1. Literature Review

There is worldwide agreement that the number of children diagnosed with Autism Spectrum Disorder (ASD) is increasing dramatically [1,2,3,4,5]. Obtaining an accurate diagnosis for children with ASD and their families is challenging, as this process has a significant influence not only on the children but also on their parents [6,7,8,9,10]. There is no doubt that an early diagnosis is particularly important for children, since it usually allows them to access timely and appropriate early intervention services and other support systems [7,11]. For parents, it increases quality of life [10,12], facilitates coming to terms with ASD [4], and brings them a sense of relief [13].

Dissatisfaction with the diagnostic process has remained a pervasive theme that is repeatedly reported by the parents of children with ASD worldwide [14,15,16,17,18,19,20,21]. This dissatisfaction seems to be caused by the long diagnostic process, the absence of clear information, miscommunication with professionals, multiple referrals, and insufficient post-treatment support [22,23,24,25]. In a recent study, Makino et al. [26] reviewed 122 articles published between 1994 and 2020 that investigated parental perceptions of the diagnostic process of ASD in high-income-country populations. Their review identified common aspects of autism assessments, for example, delays in the process, the use of invalid tools, unqualified professionals, a dissatisfactory way of delivering diagnosis results, and an insufficient support model. More recently, this argument was further supported by Small and Belluigi [27], who systematically reviewed 34 publications from 16 countries that examined parental satisfaction with ASD diagnoses. Parents expressed satisfaction with the process in only eight of the thirty-four reviewed studies. Studies involving adults with ASD have revealed similar findings, with assertions of feelings of dissatisfaction with post-diagnostic services [7,28,29,30].

However, the literature also reports that some parents express positive feelings toward the diagnostic process, for example, being informed and listened to [31], having their children’s behaviors understood [14,32], experiencing excitement as they benefited from their child’s assessment [4], receiving practical support, and feeling empowered [28]. Kanfiszer et al. [33] interviewed adult women with ASD, some of whom felt that the diagnostic process had enabled them to reconstruct their previous experiences and improve their abilities.

Findings from the Arab world on the diagnosis and evaluation of children with ASD are extremely rare. Alallawi et al. [34] reviewed 70 social, educational, and psychological studies published in English that enrolled persons with ASD and their family members from 14 Arab countries. They found that most of the research had been carried out in Jordan, the vast majority of the included studies used quantitative methods, and very few had investigated the issue of diagnosis; these studies were mainly focused on the development and validation of diagnostic tools for use in the Arab world. These results were confirmed in a more recent study, highlighting the gap in research on the diagnostic process for children with ASD in Arab countries, particularly in Jordan [35].

It has been suggested that an understanding and consideration of parental views is important for appraising the suitability of current practices for the diagnosis of ASD [26]. Although Jordan is a keystone of the research published in the field of ASD in Arab countries [34,35], the experiences of Jordanian parents of children with ASD with the diagnostic process remain unknown. To date, research investigating perceptions of the diagnostic process for children with ASD has largely been carried out in Western countries. Accordingly, this paper considers Jordanian parents’ experiences and perceptions of the process of their children being diagnosed with ASD.

### 1.2. The Current Status of Diagnosing ASD in Jordan

Comparatively little is known about the statistics relating to ASD in Jordan [36,37,38]; this may be due to cultural issues, unreliable diagnostic criteria, a dearth of related research, a lack of professionals in the field, and economic factors [34,39,40]. Although there are few studies on the prevalence of ASD in Jordan, it is estimated that one out of every 50 children has ASD, for an approximate total number of 10,000 children with ASD in Jordan [41]. This estimation is lower than the prevalence observed in the United States and European countries [3].

Although Jordan has recently implemented newborn screening tests and a comprehensive vaccination program, there is no infrastructure in place for newborns and toddlers to receive preventive medical checkups and no organized healthcare visiting service for monitoring children’s development. Therefore, there are no standardized pathways for parents to follow in order to obtain a reliable diagnosis for their children with suspected ASD. Parents usually go back and forth between professionals, beginning with pediatricians, based on their own choices and without any need for a referral. When seeking a precise diagnosis for their child, parents will typically engage with professionals from many health-related, psychological, and educational disciplines, and may keep moving between pediatricians, psychologists, psychiatrists, speech therapists, physical therapists, occupational therapists, and special educators.

Although extensive research has been conducted on the diagnostic processes for ASD worldwide, we identified no studies that examine Jordanian parents’ experiences in obtaining diagnoses for their children with ASD. Most of the extant studies were conducted in Western countries, which have very different social contexts and health practices; this may affect their diagnostic processes. Furthermore, the relevant literature has made significant use of quantitative approaches when examining the diagnostic processes for children with ASD, with fewer qualitative studies being undertaken. Accordingly, this research aimed to provide a deeper understanding of the diagnostic process for children with ASD by qualitatively examining its various aspects within the Jordanian context. Specifically, the key question addressed in this research was how Jordanian parents experienced the process of their children being diagnosed with ASD.

## 2. Materials and Methods

### 2.1. Research Design

This research aimed to generate a descriptive, detailed account of how Jordanian parents experienced the process whereby their children were diagnosed with ASD. A descriptive qualitative approach was utilized to generate a rich dataset [42,43], with descriptions and examples that put the participants’ language and concerns at the forefront [44]. A qualitative approach helped to address the unique context of Jordanian parents of children with ASD [45]. This may help to provide satisfactory information to encourage policy makers to improve the quality of diagnostic services for children with ASD in Jordan.

### 2.2. Ethical Consideration

An ethical approval form was completed and sent to the Research Ethics Committee (REC)/College of Scientific Research (CSR) at Al-Balqa Applied University. The formal research ethical approval was granted by the RET/CSR after ensuring that the requirements were met, including signed consent forms by the participants, secured anonymity for the participants, the parents being allowed to withdraw from the research whenever they wanted, and the data being kept secure and confidential.

### 2.3. Participants

The current study included 18 parents of children diagnosed with ASD. The parents were recruited based on a purposive sampling approach, which was ‘to ensure that there is a good deal of variety in the resulting sample, so that sample members differ from each other in terms of key characteristics relevant to the research question’ [42] (p. 418). Participants were selected according to one criterion—being the primary caregiver for a child diagnosed with ASD. Participants were recruited from the special education centers and schools that their children were enrolled in. This meant that it was challenging to recruit parents of children who had recently received formal diagnoses, as their children had not yet started attending special education centers or other educational settings. The first author approached the principals of these institutions and discussed the research purposes and procedures with them. He also sent the participant information sheet and consent form via the WhatsApp platform, to be forwarded to potential participants. The principals then contacted eligible parents, asking them if they wished to voluntarily take part in this research. This process resulted in 21 parents who expressed their willingness to participate. Their phone numbers were provided by the principals, and the researchers then contacted the candidate parents to set up suitable times and places to carry out the interviews. Three parents were later unable to attend their interviews. Detailed information about the participants is shown in Table 1.

### 2.4. Data Collection

A semi-structured interview was deemed the most suitable method for collecting the data. Because of its flexibility, it allowed the researcher to track the important ideas that were raised during the interviews [42,46]. Eighteen semi-structured interviews were conducted with twelve mothers and six fathers, guided by an interview schedule that was developed by extensively reviewing the literature. Table 2 provides translated examples of the proposed questions. The interviews were carried out in Arabic, which was the participants’ native language and were audio recorded, lasting for 40 to 70 min. The interviews were conducted in three different settings based on the participants’ preferences. While eleven parents preferred to be interviewed at their children’s schools, two parents preferred their homes, and five were interviewed over the phone.

### 2.5. Data Analysis

Thematic analysis, as described by Braun and Clarke [47], was deemed the best technique for the analysis of the transcript data because our aim was to provide a rich thematic description of all the collected data that would enable a reader to grasp the significant themes. According to Braun and Clarke [47], thematic analyses may be especially helpful when working with participants whose opinions about the topic being studied are unknown. This was the status of Jordanian parents of children with ASD—their voices on the process of their children being diagnosed had been overlooked. The authors independently began the analysis process by reading the transcripts repeatedly and extensively to become familiar with the data, generating initial codes by highlighting interesting patterns and significant features that pertained to the research aims, clustering and organizing the codes to generate possible themes, testing how well the themes related to the coded extracts and the entire dataset, giving each theme a name, clarifying the meaning of each theme and determining how it helped with an understanding of the parents’ accounts, and, finally, writing the research report. It is worth mentioning that the analysis process was reiterative and the authors met frequently to discuss the themes, negotiate contradictions, and reconcile the final codes and themes.

To ensure the credibility of the current research, we employed the respondent validation technique described by Bryman [42]. We emailed four participants with full transcriptions of their interviews, asking them to reply with feedback about their accounts. These participants expressed their satisfaction with the transcripts and did not suggest any changes. The process undertaken by the researchers of independently coding the data and then reaching an agreement could ultimately be considered to constitute inter-coder reliability [43,48].

## 3. Results

The analysis process identified three key themes that captured Jordanian parents’ perspectives on their journeys to obtain confirmed ASD diagnoses for their children. These themes were dissatisfaction with professionals’ ability to approach parents, the unstructured nature of the diagnostic process, and perspectives on the diagnostic tools. Figure 1 highlights the major themes and related sub-themes that encompass Jordanian parents’ perspectives on obtaining a diagnosis of ASD for their children. It should be noted that the identified themes were not viewed as being independent; rather, they overlapped and were largely connected to each other.

Table 3 clarifies the findings by providing a summary of the key points of the parental journey to obtain a diagnosis for their child. This includes information about the child’s age when the parents first had concerns, the first (main) symptoms, the first professional point of contact, the time delay between first concerns and the diagnosis, and the professional who made the final diagnosis. This information is presented as given by the parents.

### 3.1. Dissatisfaction with Professionals’ Abilities to Approach Parents

Our participants’ journeys towards seeking diagnoses began when they were alerted that their children had developmental issues. They initially sought to contact health practitioners, especially pediatricians, rather than other professionals, as they were attempting to medically determine their children’s problems and find cures. Most parents talked about their first visits to pediatricians and discussed how their concerns and fears about their children’s symptoms were assuaged by doctors. Although this practice of ‘reassurance’ was a source of relief for the parents, they felt uncertain and believed that timely diagnoses and early interventions for their children were hindered. Some parents perceived ‘reassurance’ as a sort of ignorance, whereby their parental concerns were not sufficiently addressed by the doctors. Parent 1 had noticed difficulties in her child’s communication skills when he reached the age of 30 months. She said the following about her first visit to a pediatrician: “He did not seriously consider what I said about my child’s language difficulties. I told him my child cannot say even a word well, he told me “do not worry, Faris is still young and he will speak as he grows. All the results of his medical tests are normal. Please do not be dubious”.

When parents returned to the pediatricians to reiterate their concerns, the doctors tended to refer them to other professionals. Most parents spoke about their referrals between professionals and clinicians (e.g., pediatricians, psychologists, nutritionists, neurologists, audiologists, physical therapists, occupational therapists, and special educators). The parents provided anecdotes about their experiences with those professionals and clinicians and their accounts clearly revealed many feelings, such as being emotionally unsupported, uninformed, undirected, and discarded. Parent 12 recounted the experience of being referred: “You cannot imagine how many healthcare professionals we have seen. Pediatrician, neurologist, genetic counselor, audiologist, phycologist, and speech therapist. Honestly, some of them were kind but they were not helpful. They just wanted to focus on their specialties; there was no further information about autism. Some of them prescribed medicines and others suggested a special diet. I think they just wanted to avoid saying that your child has ASD”.

The recurring referrals appeared to lead some parents to believe that the professionals lacked awareness and knowledge of ASD. This, in turn, hindered the parents taking the appropriate actions for their children’s development and prevented them from building a sense of certainty. For example, Parent 14 thought that the professionals she had dealt with were the reason for the delays in receiving timely access to early intervention services. She relayed her experience as follows: “I expected that they [the professionals] could tell me more than the word ‘autism’, give an explanation or information about where I can find services for my daughter, what educational setting is available for her. Most likely, they [the professionals] did not have such information nor awareness nor knowledge about autism. They should have advised me to enroll Leen in an appropriate training program”.

The participants identified the disclosure of the ASD diagnosis as a further unsatisfactory aspect of communication with the professionals. Some parents talked about how, when delivering the diagnosis, the professionals did not consider the immediate impacts that it would have on them. The participants tended to be dissatisfied with the way in which the professionals dealt with them at that unforgettable and sensitive moment. This was accompanied by the abovementioned feelings expressed by the parents. Parent 17 remembered the following about the occasion on which she was informed about her child’s diagnosis: “He [the psychologist] could tell me in a different way this bad news. He said it directly, “your son has autism”, without any introduction. I asked him what does that entail. He did not explain enough; he just counted what Jad could not attain in the scale items. I mean, he could encourage me by saying, for example, your son’s development will be better if you enroll him in an early intervention program and take care of him”.

Despite the fact that most of the participants commented negatively about their interactions with professionals, we found some satisfied parents. A few tended to compare their communication experiences with public and private health professionals; they were happy about how they were approached by professionals working in the private sector. Highly educated parents could read the English language, which enabled them to review reliable websites that provided information about ASD and thus obtain the knowledge necessary to discuss their children’s cases and share information with the professionals. Parent 8, who has a Ph.D., remarked: “I have read a lot about autism so I did not feel that the professional was saying something new. He said similar things to what I had expected […] I am educated enough and my English is very good [thanks God], so I looked for the information on the internet. So, I can say yes, I was convinced and satisfied […] and you know there are differences between public and private services, private service providers are highly interested in keeping you satisfied. They let you say what you want and they listen to you and reply politely, they do not ignore anything […] to be fair, I did not visit public clinics”.

### 3.2. Unstructured Diagnostic Process

Although the participants had obtained the diagnoses several years earlier, they were keen to discuss the unstructured diagnostic process that they had experienced. They commented on the time that had passed since they first identified that their children had developmental issues, and this was frequently connected to the involvement of multiple specialists. The time it took to receive a diagnosis varied among the participants. The parental estimations of the time it took to receive a final diagnosis from their initial concerns ranged from approximately one to four years. Some parents reported receiving misdiagnoses at the beginning of the diagnostic process, such as deafness, childhood depression, intellectual disability, attention deficit hyperactivity disorder (ADHD), and social disorders. Parent 9, the mother of a boy, talked about her son receiving two different misdiagnoses before he was diagnosed with ASD: “It took around three years to confirm that Nedal has ASD. You know, it is funny (laughing), actually, he had received two diagnoses before he was identified as a child with ASD. At the first visit to the clinic, Doctor X told us “he has ADHD”, then a psychologist tested him and categorized him as intellectually disabled”. 

Unlike most participants, a few parents reported that their children were diagnosed quickly and that the process did not take more than two sessions. It was notable that those parents were well-educated and had read extensively about their children’s cases before they saw clinicians; they appeared to be looking for confirmation of what they had already thought. Their attained knowledge of ASD may have helped them come to terms with the diagnosis and this, in turn, enabled them to obtain a diagnosis in a shorter timeframe. For example, Parent 18 was an educated father working in the medical field, with a master’s degree in laboratory analysis. He recounted the following experience of obtaining a diagnosis for his child: “I told the pediatrician he might have ASD. He [the doctor] replied, “I cannot confirm that. There is no blood test, brain scan, or other medical test that can diagnose autism”. I already knew that as I had checked international resources related to my child’s symptoms, my profession helped me… Accordingly, I then went to a special educational center, they assessed his cognitive, social, language, and behavioral development and they confirmed the case”.

It may be surprising that, while the children of the participants were enrolled in educational institutions and receiving special services, some parents continued to doubt the diagnosis. They were in disbelief about the diagnosis, claiming that their children had been diagnosed inaccurately. Parent 6 insisted that the diagnosis given to his daughter was unconvincing: “Yes, we got a statement from the X center confirming that she has autism. However, Farah is not an autistic girl. I know she has a developmental problem but not autism. I am sure she does not have autism. I think the way she has been diagnosed was wrong… She may have learning difficulties or slow learning, but not autism”.

Generally speaking, all participants were keen to talk about the high cost of the services offered for children with ASD, including diagnosis and intervention. However, the parents who had taken their children to private institutions to obtain a diagnosis reported that the expenses of the diagnostic process were extremely high. Those parents tended to recount how much they paid at every place they had attended, while also claiming that the services were insufficient. Parent 10 said the following: “I do not know how families of children with ASD can manage their financial issues. I am wealthy (thanks God) and I could pay for the diagnosis requirement but I wonder how others can. You know, each time we visited a specialist, we had to pay a huge amount of money. For example, the visits to the pediatrician cost me around 100 JD, audiologist 50 JD, articulation therapist 40, and do not forget the medical tests. You live here and you know how much it costs”.

### 3.3. Perspectives on Diagnostic Tools

For most participants, the final statement about whether or not their child had ASD was made by psychologists or educators who relied on diagnostic tools and observations when making their decisions. Some comments were made by the parents about the instruments used by the professionals to determine their children’s abilities and skills. They felt suspicious about the quality of the screening and diagnostic measurement tools, and talked about their capacity to effectively identify ASD. Indeed, the participants, especially the educated parents, expressed dissatisfaction with the instruments used; they criticized the results and reliability of the diagnostic tools that their children had experienced. Parent 16, the father of a six-year-old boy, expressed the following sentiments about these tools: “They applied several tests and scales on my son such as Wechsler Intelligence Scale for Children, CARS2, Autism Behavior Checklist, and Autism Severity Scale. It is not reasonable that conducting such tools can determine whether the child autistic or not. I do not think these tools are an effective way for diagnosing autism. There should be alternative methods. Do you think these instruments are able to produce an accurate diagnosis? I think some of them are outdated”.

The competence of the specialists who used the diagnostic tools was another issue that some participants raised. Parents commented on the procedures that the professionals followed when they conducted screening sessions for their children. The parents appeared to be unfamiliar with some elements of the diagnostic tools, and the professionals were unable to simplify the components of the tools. Sometimes, parents felt that, while they attended multiple sessions where the tools were administered, they did not have enough time to accurately respond to the items. For example, Parent 11 shared the following experience of a test having been administered for her son: “When the professional asked me to rate my child’s behaviors… For example, I still remember there was an item that was about my son’s response ability to visual stimulations. I could not accurately estimate which statement fit, and the professional did not help me sufficiently although I asked him to clarify. I do not think he was qualified enough; he should have had training on using that scale”.

## 4. Discussion

Reed and Osborne [49] (p.1) state that “a lack of knowledge regarding best diagnostic practice may ultimately impair treatment efficacy and lead to increased health- and economic-burdens”. It is therefore important to investigate the diagnostic process in Jordan in order to increase knowledge in this area. The main objective of this research was to investigate how Jordanian parents perceived the process of obtaining an ASD diagnosis for their children. A key finding from the analysis of interviews with Jordanian parents of children with ASD was that, although there are differences in the medical, educational, and support systems available in other parts of the world, they exhibit many similarities to their peers worldwide in terms of their journey towards receiving a diagnosis. They appeared to be dissatisfied with the diagnostic services available in Jordan, reporting that the professionals were unable to approach them properly. Additionally, the parents described several aspects of the unstructured diagnostic process and commented on the diagnostic tools used.

### 4.1. Dissatisfaction with Professionals’ Abilities to Approach Parents

It is well known that the diagnosis of ASD is a complicated process that involves professionals from several different disciplines, including health, education, and psychology [13,19,50,51]. For our participants, the medical path was the initial track they followed to understand their children’s cases, which is similar to the experience of other parents of children with ASD worldwide, for example, in the UK [6], in Germany [17], in France [52], in New Zealand [15], and in Hong Kong [53]. From the perspective of Jordanian parents of children with ASD, the pediatricians downplayed their concerns about their children’s development, a finding also reported in the study conducted by Hidalgo et al. [23], where 80% of the participants perceived pediatricians as the reason for the delay in their child’s diagnosis. This was usually associated with unfruitful communication and feelings of being unsupported, uninformed, and undirected [22,26,27].

The accessibility of diagnostic services for children with ASD in Jordan entails several referrals, as was evident from previous findings [7,24,25]. Jones et al. [30] found that the processes for obtaining diagnoses for their respondents required three to six referrals. An earlier study conducted by Goin-Kochel et al. [18] found that visiting fewer professionals increased parental satisfaction with the diagnostic process. This idea was echoed in the results of our investigation—our participants visited many clinicians and, accordingly, seemed dissatisfied with their journey towards a diagnosis.

The way in which an ASD diagnosis is delivered is critical to the parents [26], and it influences their later coping strategies [54,55]. For parents of children with ASD, being satisfied with the disclosure of an ASD diagnosis is strongly connected to receiving adequate information [56,57]. This can also be applied to the Jordanian parents of children with ASD who expressed their dissatisfaction with the professionals’ methods of communicating the diagnoses—they were informed in a poor manner and were left unsupported and lacking information.

Taking into account the local cultural context, parents being dissatisfied with the professionals’ performance and visiting different specialists could mean that they were looking for a professional who could tell them that their child did not have ASD. This is especially true where a diagnosis of ASD is likely to be understood as attaching a stigma to the family and shattering their identity as normal parents [58,59]. ASD is perceived as belonging to a medical model and entails numerous challenges for parents, including being blamed for their children’s disorder, bearing a constant burden, and facing the absence of reliable education systems for children with ASD [36,40]. Jacobs et al. [60] (p.11) state that “in medical practice, a particular diagnosis is viewed as leading to a particular cause and cure”. We expected that this perception was shared by our participants; the professionals did not prescribe a cure for their children, which in turn led the parents to be dissatisfied with the professionals.

### 4.2. Unstructured Diagnostic Process

The multiple referrals that our participants experienced led to delays in obtaining confirmed diagnoses, and this experience is not unique to Jordanian parents of children with ASD. This aspect of the diagnostic process has been extensively reported in the literature [15,16,26,27,61], and it ultimately results in delays in accessing early intervention services, hindering children in reaching their full potential [9,11,50].

Some parents reported receiving misdiagnoses at the beginning of the diagnostic process. A possible explanation for this is that the process of diagnosing ASD is complicated [27,51,62] as ASD is heterogeneous and involves symptoms that overlap with those of other developmental disorders. Furthermore, based on the notion that the negative attitudes towards ASD in Jordan are more pronounced than those towards other disabilities [59], it could be assumed that the professionals were fully aware of the social stigma that is inevitably attached to ASD in Jordan; they may, therefore, have tried to alleviate this potential burden by suggesting less stigmatized labels, for example, hearing impairment, learning difficulties, or ADHD. This finding is in line with the results of the study conducted by Fusar-Poli et al. [63], which found that approximately two-thirds of their participants had been diagnosed with one or more conditions before they were confirmed as having ASD.

Although our participants had received confirmed diagnoses, some still had doubts and were negotiating their children’s cases. This may have been due to their dissatisfaction with different aspects of the diagnostic process. On the other hand, this perspective could be seen as a form of denial that the parents were experiencing [14,25,26,31], whereby they rejected the fact that their children had been labeled as having ASD. Farrugia [64] found that, when a child is identified as having ASD, their parents may start to fight feelings of stigmatization. This was evident among some of our participants; they defended themselves from the stigma by trying to place their children in other classifications of disabilities that entailed less stigmatization.

Leslie and Martin [65] (p. 354) concluded that “it is clear that health care expenditures for children and adolescents with autism are considerable and are increasing over time”. This was confirmed by the parents’ stories in this study; they repeatedly recounted how much they had paid for the services provided to their children, including the diagnostic services. This was especially true for the parents who had visited private health clinicians. Similarly, the expense of ASD assessments was perceived as a challenge to diagnosing ASD by Latino parents in the United States [66]. Caucasian and African American families in Ohio experienced similar barriers when they sought diagnoses for their children with ASD [25].

### 4.3. Perspectives on Diagnostic Tools

For most participants, the last step in the diagnostic process was the administration of diagnostic assessment tools, and the final diagnosis was made based on the results of these assessments. While diagnostic tools may currently be the most common and useful means of demonstrating ASD [20,67], the parents in this study commented in various ways on the assessments undertaken by their children. These comments included the proficiency of the professionals who administered the assessment tools and their workability, and relate to the features of diagnostic tools proposed by Mayes et al. [68], such as being easily achievable, affordable, validated, inexpensive, short, simple, practical to administer, accurate when filled out by parents and clinicians, and suitable for different severity levels and ages. Similarly, Hanratty et al. [69] reviewed six instruments measuring characteristics used to assess behavioral symptoms in children with ASD, and their review revealed the tools’ shortcomings in terms of their reliability, validity, and capacity to trigger interventions. This may not be consistent with the views of professionals who are actively enrolled in diagnosing ASD in the UK [70], where 75% of them rated the diagnostic tools as very or quite helpful.

In previous studies, parents have consistently expressed that they felt frustrated and dissatisfied with the instruments used to identify children with ASD. For example, in a study conducted by Crane et al. [7], the researchers asked parents of children with ASD to provide feedback on the assessment tools utilized in the diagnostic process. The parents commented on the language of the instruments and on the environment in which the tools were used. Similarly, our participants reported difficulties in fully understanding the diagnostic tools when they were asked to rate their children’s symptoms. Professionals were unable to present the instrument items simply and easily to the parents. Additionally, when a diagnostic tool is being implemented, a child can be described as a third party in the process; the symptoms of ASD are passively observed and assessed by professionals or parents, and it is rare to collect the data directly from the child who is being diagnosed, which may result in unfairness or misunderstandings [71].

The Higher Council for the Rights of Persons with Disabilities (HCRPD) has tried to develop and improve the assessment and diagnosis process in Jordan by taking serious steps, such as the adoption of credible and validated criteria that contribute to enhancing diagnosis outcomes, training professionals throughout the country, and ratifying an agreement with the Ministry of Health and Health Care Accreditation Council to develop evidence-based clinical practices that will ultimately unify the diagnosis procedures for children with ASD and intellectual disabilities in Jordan [72]. Despite the recent attempts by the HCRPD to incorporate these steps, the diagnostic process for children with ASD in Jordan is still unsystematic and disorganized. The HCRPD [72] stated that several challenges were encountered when attempting to enhance the diagnosis process for children with ASD in Jordan, including the following: (1) various methods were used by the diagnosis centers; (2) the diagnostic centers lacked accurate tools and equipment; (3) a range of diagnosis forms were used by the diagnosis centers; (4) there was no sponsored budget for the diagnosis centers; and (5) insufficient time was taken to diagnose ASD.

Although this research provided a space in which the opinions of Jordanian parents regarding the diagnostic process can be heard, it has several limitations that should be addressed. First, the sample size was somewhat small; this is a typical drawback of qualitative studies, which can be overcome by conducting additional research to validate the initial study’s findings. A further limitation is that the accuracy of some participants’ reports of their diagnostic process may have been impacted by the time that passed between these diagnoses and the interviews. Additionally, the length of time between the appearance of symptoms and the diagnosis may potentially restrict the generalizability of the study results, since the situations could have changed in the time that followed. Finally, the participants in this study agreed to take part in this research voluntarily, so they may have had different perceptions than other parents of children with ASD across the country.

## 5. Conclusions

While the interviews captured crucial events and feelings, the participants also expressed their happiness at being able to tell their stories. This raises an important implication for practice, which is that parents need their voices to be heard. Our findings suggest that healthcare professionals should receive competency training on how to interact effectively with parents. This can result in the development of stronger relationships, enabling practitioners to offer effective support, which ultimately facilitates patients’ access to early intervention services. The adoption of reliable and validated criteria that help to improve diagnosis outcomes, training professionals across the country, and strengthening coordination among the multidisciplinary teams involved in the diagnostic process can improve the experiences of parents of children with ASD.

## Figures and Tables

**Figure 1 children-10-01394-f001:**
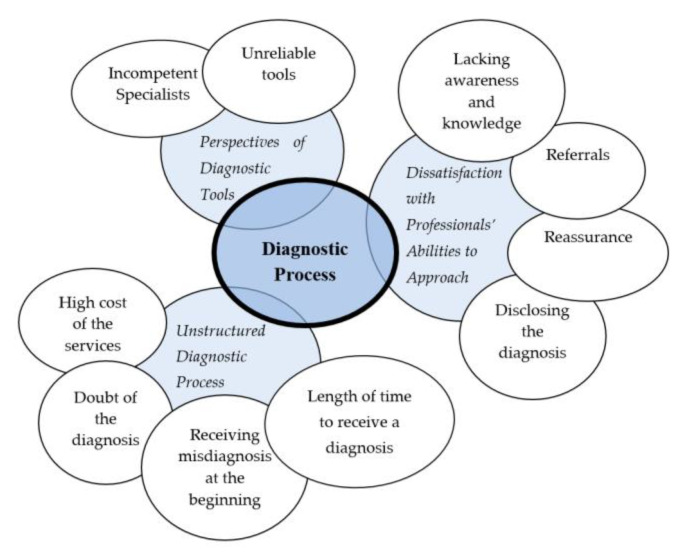
Major themes and related sub-themes.

**Table 1 children-10-01394-t001:** Demographic characteristics of the participants.

Parent					Child with ASD			
No.	Age	Education	Monthly Income JOD	Urban/Rural	Pseudonym	Age	Gender	Functional Level
1	45	Bachelor	600	Urban	Sameer	7	Boy	High
2	40	Bachelor	650	Urban	Rami	8	Girl	High
3	39	Bachelor	Not given	Urban	Faris	8	Boy	High
4	37	Diploma	600	Urban	Laith	7	Boy	High
5	45	Bachelor	1100	Urban	Oon	10	Boy	Middle
6	42	High School	500	Urban	Farah	6	Girl	High
7	47	Diploma	450	Urban	Naser	10	Boy	High
8	42	Ph.D.	1600	Urban	Nashmi	7	Boy	Middle
9	30	Bachelor	800	Urban	Nedal	6	Boy	Middle
10	45	Bachelor	600	Rural	Odai	7	Boy	Middle
11	31	Bachelor	400	Urban	Saad	6	Boy	Low
12	45	Diploma	650	Urban	Bader	9	Boy	High
13	44	High school	700	Urban	Majd	11	Boy	High
14	44	High school	Not given	Rural	Leen	9	Girl	High
15	38	Bachelor	600	Urban	Reem	7	Girl	High
16	37	Masters	1300	Urban	Baram	6	Boy	Low
17	30	Bachelor	1000	Urban	Jad	5	Boy	High
18	44	Masters	1500	Urban	Zaid	11	Boy	Middle

**Table 2 children-10-01394-t002:** Examples of the proposed questions.

Can you tell me about the diagnostic process for your child, starting from the beginning of the journey? What symptoms had you seen in your child that encouraged you to seek a diagnosis? How old was your child when you first became concerned about his or her development? How old was he or she when you received the final diagnosis? Can you tell me about the communications with the professionals with whom you have dealt with throughout your journey of seeking a diagnosis for your child? What do you think about the final diagnosis report that you received? Do you think it is easy to obtain an accurate diagnosis for children with ASD in Jordan?

**Table 3 children-10-01394-t003:** Summary of the key points of the parental journey to obtain diagnoses.

Child’s Pseudonym	Age at First Parental Concerns	First Symptoms	First Professional Contact	Time Delay between the First Concerns and the Diagnosis	Who Confirmed the Final Diagnosis
1. Sameer	30 months	Language delay	Pediatrician	30 months	Psychologist
2. Rami	36 months	Avoiding eye contact	Pediatrician	24 months	Psychologist
3. Faris	Did not recall	Not responding to his name	Pediatrician	-	Special education teacher
4. Laith	36 months	Stereotyped behaviors	GP	20 months	Special education teacher
5. Oon	Did not recall	Language delay	Pediatrician	-	Special education teacher
6. Farah	24 months	Stereotyped behaviors	Pediatrician	36 months	Psychologist
7. Naser	40 months	Language delay	GP	48 months	Special education teacher
8. Nashmi	30 months	Avoiding eye contact	Pediatrician	12 months	Special education teacher
9. Nedal	34 months	Hyperactivity	Pediatrician	24 months	Special education teacher
10. Odai	24 months	Language delay	Pediatrician	30 months	Psychologist
11. Saad	26 months	Not responding to his name	Pediatrician	18 months	Psychologist
12. Bader	38 months	Stereotyped behaviors	Pediatrician	34 months	Special education teacher
13. Majd	Did not recall	Not responding to his name	Pediatrician	-	Psychologist
14. Leen	Did not recall	Language delay	Pediatrician	-	Psychologist
15. Reem	36	Avoiding eye contact	Pediatrician	20 months	Special education teacher
16. Baram	32 months	Language delay	GP	24 months	Psychologist
17. Jad	36 months	Language delay	Pediatrician	20 months	Psychologist
18. Zaid	36 months	Language delay	Pediatrician	14 months	Psychologist

## Data Availability

Data are available from the corresponding author upon reasonable request.

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
