# Peer review of "The Diagnostic Process for Children with Autism Spectrum Disorder: A Preliminary Study of Jordanian Parents’ Perspectives"

_children, 2023, doi:10.3390/children10081394_

Round 1

Reviewer 1 Report

Thanks for the opportunity to review your manuscript. Parental perspectives on clinical procedures are essential to improve child and family outcomes of early intervention.

1. The introduction is clear and compact.  in section 1.2. I would like to get more information about the current situation of autism diagnosis in Jordan. Is there a system of preventive medical check-ups for infants and toddlers, eg. by pediatricians, GPs, visiting nurses? Are autism screeners used (universally, in specific cases ...), have trainings been provided for the screeners? Give information about who makes the diagnosis (pediatrician, psychology, multi-professional centres...?) and about the usual ways of referrals.

2. Participants: Please state explicitly the criteria (aiming for high diversity of the sample): mainly child-related (gender, current age, severity of autism, nonverbal and verbal development, behavior problems....) or also family related (urban, education, income....)?

3. The time-lag between the parent interviews and the time of ASD diagnosis for their child looks significant. Can you give more information explaining this latency (eg very late diagnoses, limited acces to parents of children who were recently diagnosed...).

4. You might decide for coding your particpants (number code) instead of labelling by names. Consider a principle of order in table 1 (eg by parental education and age) ...

5.  Results: Before presenting your results I would like to get more information for each child (and the whole group) about eg. the child's age at first parental concerns, first symptoms, age at final diagnosis, time delay between first concerns and the diagnosis, first professional point of contact, who made the final diagnosis ....! This information appears critical, as it might explain a great deal of the parental dissatisfaction with the process. I recommend to present this information by preparing a table (in the beginning of the results section).

6. 3.2. Please give a more specific theme. "Negative aspects of the Diagnostic Process" is very vague. In addition, the other two themes are also negative! Is this section more about the time-delay related to the lack of a specified procedure for ASD identification in Jordan?

7. limitations line 478: Please mention that the time delay before diagnosis might also limit the generalizabiliy of your results as the situation might have changed in the meanwhile.

8. conclusion: I recommend to restrict the conclusions to the paragraph beginning with line 496 and to integrate the beginning of your conclusion into the discussion.

I recommend extensive English language editing!

Reviewer 2 Report

I want to congratulate the authors for the work and the article. The study, apart from the limitations they acknowledge, provides important and valuable information about parents’ perspective on diagnostic process of ASD.

I find the manuscript is clear, well structured, results sound, and the discussion addresses the main results obtained.

The only, small, advice regarding the possibility to better summarize results to facilitate the reading. It could be useful to add a figure for highlighting the main findings.

Reviewer 3 Report

Dear Authors,

I am very pleased to be a reviewer of this clinically important work. As a reviewer, I would like to ask the authors' team to respond to the suggestions below: 

1. Structuring the introduction of the work and limiting it to aspects relevant to the introduction to the work: presenting the main information about autism, including in terms of aspects related to the occurrence in Jordan, presenting issues related to parents of children from the perspective of the diagnostic process (issues discussed so far in the literature) , presentation of the main information about the diagnostic process in Jordan. Moving the remaining aspects to discussion.

2. I have doubts as to whether the 18 parents, including 12 mothers and 6 fathers, are a representative group - I would suggest increasing the size of the study group or changing the title to a preliminary report and then continuing work on the subcjet. 

3. After increasing the size of the study group, a significant comparison can be made in terms of comparing the results between mothers and fathers of children with ASD (or in future studies). 

4. It would be worth presenting the research results in a more readable way.

5. Some excerpts presented in the introduction should be brought into the discussion

6. In conclusions, it would be good to focus on those statements that result from the research results.

7. The limitations section should be entered.

Kind regards,

Reviewer

Round 2

Reviewer 3 Report

Dear Authors,

Thank you for considering my suggestions - I hope they were helpful.
I wish you fruitful further research.

Kind regards,
Reviewer